# CLUSTER-BASED FEATURE IMPORTANCE LEARNING FOR ELECTRONIC HEALTH RECORD TIME-SERIES

## ABSTRACT

The recent availability of Electronic Health Records (EHR) has allowed for the development of algorithms predicting inpatient risk of deterioration and trajectory evolution. However, prediction of disease progression with EHR is challenging since these data are sparse, heterogeneous, multi-dimensional, and multi-modal time-series. As such, clustering is used to identify similar groups within the patient cohort to improve prediction. Current models have shown some success in obtaining cluster representation of patient trajectories, however, they i) fail to obtain clinical interpretability for each cluster, and ii) struggle to learn meaningful cluster numbers in the context of the imbalanced distribution of disease outcomes. We propose a supervised deep learning model to cluster EHR data based on the identification of clinically understandable phenotypes with regard to both outcome prediction and patient trajectory. We introduce novel loss functions to address the problems of class imbalance and cluster collapse, and furthermore propose a feature-time attention mechanism to identify cluster-based phenotype importance across time and feature dimensions. We tested our model in over 100,000 unique trajectories from hospitalised patients with Type-II respiratory failure to predict four different outcomes. Our model yielded added interpretability to cluster formation and outperformed benchmarks by at least 5% in mean AUROC.

## 1 INTRODUCTION

Chronic conditions such as Chronic Obstructive Pulmonary Disease (COPD) and Cardiovascular Disease (CVD) describe a broad spectrum of medical ailments, and affect a significant percentage of the overall population (Adeloye et al., 2015). Such diseases are characterized by the existence of multiple distinct patient subgroups, largely distinguished by differences in pathology and in the response to different treatments and medical interventions Turner et al. (2015); Vogelmeier et al. (2018). Exacerbation of COPD, a condition of respiratory failure, can result in emergency hospital admission and mortality if it is not well treated and managed. Early identification of COPD patients' subgroups is therefore of high medical importance and relevance. EHR time-series data are typically used to determine clinically relevant COPD inpatient subgroups, and have been applied to detect risk of deterioration (Pikoula et al., 2019).

However, modelling disease progression and risk prediction is challenging due to the extreme data heterogeneity nature of EHRs. Firstly, EHR data contains a mixture of demographic or static variables (i.e. time independent such as age and sex), and multi-dimensional time-series (e.g Heart Rate, HR, and laboratory measurements, such as blood tests). Secondly, EHR time-series are multi-modal as different features are collected from different devices, representing distinct clinical properties of relevance. Similarly, time-series features are sampled at different times and have low and distinct sampling rates, as well as different missing value properties. Furthermore, each feature is associated with different noise and evolution patterns.

Recent advances in deep learning (DL) approaches have shown promising results in EHR modelling due to their capacity to handle complex data (Rajkomar et al., 2018). Nonetheless, DL approaches lack relevant interpretability frameworks to be scaled and applied in hospital settings. Several such models have since been proposed to tackle this issue (Mayhew et al., 2018), however, most of them focus on a subset of EHR features (usually vital signs only) and fail to provide a clinically-focused phenotypic analysis of learnt patient sub-groups (via clustering).

This work builds on previous research in literature to introduce a cluster-based feature-time attention mechanism to predict patient outcomes based on EHR data. Our method also leverages phenotypic information to aid in clinical interpretability, not only making use of demographics and vital-signs information but also of relevant laboratory measurements (all present in the EHR) to provide a more complete patient physiological status. Our contributions include the following:

- An end-to-end DL supervised model to cluster EHR patient data based on the identification of clinically understandable cluster phenotypes with regard to both outcome prediction and patient trajectory in a multi-class setting;

- A weighted loss to address data imbalance for both tasks of clustering and prediction, a common issue in the medical domain;

- The incorporation of a novel loss mechanism in the model, to address the issue of cluster collapse and promote sample assignment to all available clusters;

- Finally, the inclusion of a novel interpretability framework, derived from a cluster-based feature-time attention layer, aiming to identify relevant timestamps and feature variables to represent the patient physiology, cluster assignment and, ultimately, outcome prediction.

This paper is structured as follows. In Section 2, we describe previous research in EHR time-series modelling, clustering and attention methods. Section 3 introduces the dataset used for analysis and provides description of the proposed model. The experimental setup and results of our analysis are presented in Section 4 and discussion takes place in Section 5. Finally, concluding remarks and future work are available in Section 6.

## 2 RELATED WORK

EHR data comprise complex time-series data, being high-dimensional, multi-modal and heterogeneous, and thus presenting challenges when used in machine learning models (Keogh & Kasetty (2003); Rani & Sikka (2012)). An important goal in a medical setting is to identify phenotypically separable clusters with distinct phenotypic profiles (which we denote as phenotypic clustering hereafter). For the purpose of this work, cluster phenotypes result from the combination of two distinct components: a) the evolution profile of patient trajectories' within the cluster, and b) the characterisation of the cluster with regards to clinical variables of interest. The latter may include features not used for clustering and may provide information about the underlying or future health status. Traditional clustering models such as K-Means or hierarchical clustering have been shown to fail to capture the existing time-dependent feature relationships. As such, variants have been proposed to mitigate this problem. A temporal version of the K-Means algorithm, Time-Series K-Means (TSKM, Tavenard et al. (2020)), models the distance between time-series of different datapoints, using the Euclidean distance (which is equivalent to considering all temporal observations as an independent feature value for the corresponding patient admission), or time-series alignment strategies such as Dynamic-Time Warping (DTW, Berndt & Clifford (1994)) and soft-DTW (Cuturi & Blondel (2017)).

Recent DL architectures, ranging from Auto-Encoders (AE, Ma et al. (2019)), Convolutional Neural Networks (CNN, Munir et al. (2018)) and others, have shown great promise when applied to time-series data across a variety of domains. Fortuin et al. (2019) proposed a Self-Organising Map - Variational Auto-Encoder (SOM-VAE), is a state-of-the-art, unsupervised, DL clustering algorithm which extends a variational auto-encoder architecture (Kingma & Welling, 2013) for observation learning and representation, through the addition of a Markov model (Gagniuc, 2017), to infer temporal evolution within the latent space. Clustering is performed in the low dimensional latent space through the use of self-organising maps (Kohonen, 1982) to obtain a discrete, topologically-interpretable latent representation of the learnt clusters. In a supervised setting, AC-TPC (Lee & Van Der Schaar, 2020) serves as the current state-of-the-art for identifying phenotypically separable clusters in patient trajectories in EHR data. AC-TPC maps EHR data into a latent space via an encoder, and uses an actor-critic network (Konda & Tsitsiklis, 2000) which leverages clinical outcomes to aid in cluster formation and obtaining cluster phenotypes. Neither SOM-VAE and AC-TPC provide clinically meaningful interpretation of feature-time importance or outcome of interest.

Attention mechanisms have recently been proposed to provide greater interpretability to Recurrent Neural Networks (RNN) and to aid in dealing with long-term dependencies (Vaswani et al., 2017; Xu et al., 2015), and have also been used in modelling EHR time-series (Schwab et al., 2017; Shashikumar et al., 2018). RETAIN (Choi et al., 2016) proposes a two-level reverse attention mechanism to mimic physician's decision process and predict a future diagnosis. In other recent works, attention mechanisms based on bi-directional RNN and CNN outperformed standard classification models in predicting high risk vascular diseases with the addition of medication information as input data (Kim et al., 2017). A drawback of such attention mechanisms is the focus on temporal interpretability only, and inability to look at individual features, which is key in a medical setting. To solve this issue, Shamout et al. (2019) considered independent RNN per feature, with a concatenation of the resulting latent vectors. However, the latter does not allow the joint modelling across both feature and time dimensions. Alternatively, Kaji et al. (2019); Gandin et al. (2021) proposed learning attention weights directly on the original inputs, prior to being transformed by a RNN, which does not allow modelling of the resulting latent representations. To the best of our knowledge, no existing models have been proposed that jointly leverage both feature and time dimensions (feature-time) to determine clinical observation relevance on clustered EHR data.

## 3 METHODS

### 3.1 DATASET AND PRE-PROCESSING

Our dataset was retrieved from a retrospective database of routinely collected observations from concluded hospital admissions between March 2014 and March 2018 (the HAVEN project, REC reference: 16/SC/0264 and Confidential Advisory Group reference 08/02/1394). The database includes EHR measurements of adult patients admitted to four hospitals from the Oxford University Hospitals NHS Foundation Trust. Note that the HAVEN dataset does not include data from Intensive Care Units (ICU), and we have excluded observations taken in the Emergency Department. Key characteristics of HAVEN cohort data include a) heterogeneity, b) multi-modality, and difference in: c) noise distributions, d) sampling rates, e) missing values, etc. Such properties are common across EHR settings, and are challenging with respect to learning useful representations and predictions.

We used the protocol defined in Pimentel et al. (2019) to subset the cohort to those patients at risk of developing Type-II Respiratory Failure (T2RF) in hospital (a diagram of the data selection steps can be found in Figure A.1 in the Appendix). Four patient outcomes were considered in our analysis: i) no event during hospital stay, leading to successful discharge from the hospital, or the first instance of one of three possible events, ii) unplanned entry to ICU, iii) cardiac arrest (also named "Cardiac" hereafter) and iv) death. Outcome groups are not clearly separable (see Tables A.2, A.3 in the Appendix), so patient clusters will naturally contain a mix of different admission outcomes. In this setting, the clinically relevant component of a cluster phenotype (henceforth denoted as *cluster outcome propensity* or *cluster outcome*) is represented as a categorical distribution indicating the corresponding propensity for cluster-assigned patients to each corresponding outcome.

For each admission, observations were grouped according to mean window observation value into 4 hour blocks based on the time to outcome (discharge in the case of no event during stay) - only observations within 24 and 72 hours before the outcome were considered. This time window was selected based on those traditionally used for validating Early Warning Score (EWS) systems (baseline models used by UK NHS staff to track inpatient physiology, (Royal College of Physicians, 2017) and clinical input, such that the target phenotype represents the patient status in the subsequent 24 hours. Features were transformed according to min-max normalization due to skewness and heterogeneity in their distributions. Patient admissions were randomly split into train, validation and test sets. Missing values were imputed based on the previously observed time block - all remaining missing observations were imputed according to the feature median from the aggregated validation and test data (see Section 4 for description of train-test data split). Imputed values were flagged in a three-dimensional mask matrix.

After processing, input data contained over 100,000 unique patient trajectories corresponding to 4,266 unique patient admissions (only last patients admissions were considered in our analysis). Original trajectories for the patient cohort are shown in the Appendix in Figures A.4, A.5, A.6 for different variables/features. A lack of clear outcome group separability can be observed across

temporal and static variables. Furthermore, we note the high degree of imbalance in the data - admissions with no event account for over 86.8% of the total number of admissions, while event classes correspond to 10.3% Death, 1.8% ICU and 1.1% Cardiac.

## 3.2 PROPOSED MODEL

We propose a novel model, which we denote by **C**luster-b**A**sed i**M**portanc**E** **L**earning f**O**r **T**ime-series (CAMELOT). Our proposed methodology is displayed in Figure 1[1]. Our model builds on previous literature on 3 key items: a) a modified loss function to target the multi-class imbalance, b) a novel loss function to ensure cluster assignment and phenotype are representative, and c) a novel feature-time attention-level framework to boost representation and introduce feature-time interpretability for cluster assignment.

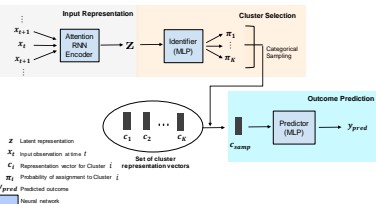

Figure 1: Diagram of proposed model. MLP - Multilayer Perceptron neural network blocks; RNN - Recurrent Neural Network.

Let $N$ denote the number of patients and $D_f$ the number of input features. Input data consists of a set of patient trajectories $\mathbb{X} = \{\{\mathbf{x}_{n,t}\}_{t=1}^{T_n}\}_{n=1}^{N}$, where $T_n$ is the maximum number of temporal observations for patient $n$, and a set of patient outcomes $\mathbb{Y} = \{\mathbf{y}_n\}_{n=1}^{N}$. Input trajectory data for the $n$-th patient is represented as $\mathbf{X}_n = [\mathbf{x}_{n,1}, ..., \mathbf{x}_{n,T_n}]$, where each $\mathbf{x}_{n,t} \in \mathbb{R}^{D_f}$ is referred to as an *observation* (vector), with a maximum of observed $D_f$ feature values. The corresponding patient outcome is a *one-hot encoded* vector $\mathbf{y}_n \in \mathbb{R}^4$ (more generally, the dimension of $\mathbf{y}_n$ equals to the number of possible outcomes).

Our DL model can be decomposed into 3 neural network blocks: an Encoder, Identifier and Predictor. We refer the action of each network respectively as E, I and P (for example, $I(\mathbf{x})$ denotes the output of the Identifier given some input vector $\mathbf{x}$). Both the Identifier and Predictor are Multilayer Perceptrons (MLP), networks of stacked feed-forward dense layers. On the other hand, the Encoder block can be further sub-divided into a) a stack of RNN layers and b) our proposed custom attention layer (see Section 3.3 for further details). Separately, we also consider a set of *trainable* cluster representation vectors, $\mathcal{C} = \{\mathbf{c}_1, ..., \mathbf{c}_K\}$. We assign the outcome for cluster $i$ as $P(\mathbf{c}_i)$.

A model call is as follows: Given the $n$-th patient input trajectory data $\mathbf{X}_n$, the Encoder network returns a *latent representation* $\mathbf{z}_n := E(\mathbf{X}_n) \in \mathbb{R}^l$. Consequently, the Identifier network computes cluster assignment probabilities, $\boldsymbol{\pi}_n := I(\mathbf{z}_n) \in \mathbb{R}^K$. Each element of $\boldsymbol{\pi}_n$, $\pi_n^i$, represents the probability assignment of $\mathbf{z}_n$ to cluster $i$, given a total of $K$ clusters. A cluster, $k_{\text{samp}}^n$ is selected according to categorical sampling ($k_{\text{samp}}^n \sim \text{Cat}(\boldsymbol{\pi}_n)$), and the corresponding cluster representation, $\mathbf{c}_{\text{samp}}^{\mathbf{n}} := \mathbf{c}_{\mathbf{k}_{\text{samp}}^{\mathbf{n}}}$ is then selected from $\mathcal{C}$. The output of the model is $y_{\text{pred}} := P(\mathbf{c}_{\text{samp}}^n) \in \mathbb{R}^4$. We note that $k_{\text{samp}}^n$ is only sampled during a *training phase*; at prediction stage, cluster selection follows the equation $k_{\text{pred}}^n = \underset{i=1,...,K}{\arg\max} \ \pi_n^i$.

## 3.3 ENCODER NETWORK AND A CUSTOM ATTENTION LAYER

The diagram of our proposed Encoder network is presented in Figure 2. The Encoder contains (i) a Recurrent Neural Network (RNN) block of stacked Long Short-Term Memory (LSTM) layers, and (ii) a customised attention layer, which computes a latent representation by comparing input data with the sequence of output states from the RNN block. We use the same notation as above,

---

[1]code will be shared online after review

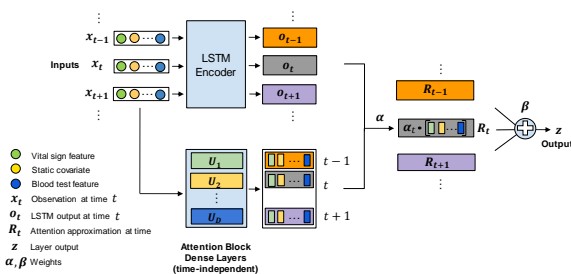

Figure 2: Diagram of the Encoder network composed of an LSTM Encoder and a custom attention layer.

and write the sequence of output states of the final LSTM layer as $\mathbf{o}_{n,1}, ..., \mathbf{o}_{n,T_n}$, with $\mathbf{o}_{n,i} \in \mathbb{R}^l$. Theoretically, each $\mathbf{o}_{n,t}$ corresponds to a representative summary of input patient information up until time $t$. We propose to approximate $\mathbf{o}_{n,t}$ as a linear combination of latent representations of each individual feature, thereby allowing the separation of output states into contributions from each feature. Note that it is important feature transformations be *time-independent*, to avoid over-parametrising the model, over-fitting and ensure feature representation maps are similar across time.

Our attention layer behaves as a set of $D_f$ feed-forward neural network layers, $\mathbf{U}_1, ..., \mathbf{U}_{D_f}$, jointly represented by: (i) a matrix of *learnable* kernel weights $\boldsymbol{D} \in \mathbb{R}_{l \times D_f}$. We write $\boldsymbol{D} = [\mathbf{D}_1, ..., \mathbf{D}_{D_f}]$; (ii) a matrix of *learnable* bias vectors $\boldsymbol{B} \in \mathbb{R}_{l \times D_f}$. Similarly, we can write $\boldsymbol{B} = [\mathbf{B}_1, ..., \mathbf{B}_{D_f}]$; and (iii) an activation function, $\sigma$ which matches the output activation of the RNN block.

Input data for patient $n$, $\mathbf{X}_n$ is fed as input to the RNN block, which outputs a sequence of latent output states $(\mathbf{o}_{n,t})_{t=1}^{T_n}$. For $t = 1, ..., T_n$, we compute $D_f$ feature representations in latent space as:

$$\boldsymbol{R}_{n,t} := \sigma(\boldsymbol{D} \odot \mathbf{x}_{n,t} + \boldsymbol{B}) \tag{1}$$

where $\boldsymbol{R}_{n,t} = [\mathbf{R}_{n,t}^1, ..., \mathbf{R}_{n,t}^{D_f}]$ is our collection of feature representations, $\sigma$ is applied element-wise and $\mathbf{A} := \boldsymbol{D} \odot \mathbf{x}_{n,t}$ is a matrix satisfying $A_{i,j} = \boldsymbol{D}_{i,j}(\mathbf{x}_{n,t})_j$. Equivalently, $\mathbf{R}_{n,t}^i$ is the output of a dense layer, $\mathbf{U}_i$ with kernel $\mathbf{D}_i$, bias $\mathbf{B}_i$, activation $\sigma$ and input $(\mathbf{x}_{\mathbf{n,t}})_i$. We approximate $\mathbf{o}_{n,t} \approx \sum_{i=1}^{D_f} \alpha_t^i \mathbf{R}_{n,t}^i = \boldsymbol{R}_{n,t} \boldsymbol{\alpha}_t$. This approximation is minimised following a least squares criterion, which has a well-known solution, $\widehat{\boldsymbol{\alpha}}_t$, and corresponding optimal approximation $\widehat{\mathbf{o}}_{n,t}$.

Given, $\widehat{\mathbf{o}}_{n,t}$, a similar procedure is used to compute a context vector as $\mathbf{z} := \sum_t \beta_t \widehat{\mathbf{o}}_{n,t}$, where weights $\boldsymbol{\beta}$ are learned to provide a more representative context vector.

### 3.4 ATTENTION MAP VISUALISATION

Given cluster representation vectors, $\mathbf{c}_k$, we can compute a cluster-wise feature-time attention visualisation map as follows. First, we normalise feature-weights $\widehat{\boldsymbol{\alpha}}_t$ according to a softmax function, $\mathbf{s}_t = \sigma(\widehat{\boldsymbol{\alpha}}_t) \in \mathbb{R}^{D_f}$, where $\sigma$ is the softmax function: $\sigma(\mathbf{x}) = \frac{\exp|\mathbf{x}|}{\|\exp|\mathbf{x}|\|_1}$. Secondly, we compute cluster-wise weights, $\gamma_t^k$ according to a least-square approximation of $\mathbf{c}_k \approx \sum_{t=1}^{T_n} \widehat{\mathbf{o}}_{n,t} \gamma_{n,t}^k$, and solved as before. We similarly normalise $\gamma_{n,t}^k$ to obtain cluster temporal scores, $e_{n,t}^k = \sigma(\gamma_{n,t}^k)$. Finally, we can compute $K$ scoring matrices, $M_n^1, ..., M_n^K \in \mathbb{R}_{T_n \times D_f}$: $\left(M_n^k\right)_{t,f} = e_{n,t}^k \boldsymbol{s}_t^f$. Note that: $\|M_n^k\|_1 = \sum_t e_{n,t}^k \sum_f \boldsymbol{s}_t^f = \sum_t e_{n,t}^k = 1$. Given that Matrices $M_n^k$ are normalised, they may be consequently, visualised as a normalised feature-time map for cluster assignment relevance and provide further model interpretability.

## 3.5 Loss optimisation

The model is optimised through consideration of three distinct loss functions. We introduce a weighted cross-entropy loss function:

$$L_{\text{pred}}(y_{\text{true}}, y_{\text{pred}}) = -\sum_{c=1}^{C} w_c y_{\text{true}}^c \log\left(y_{\text{pred}}^c\right) = -w_{c'} \log(y_{\text{pred}})_{c'} \tag{2}$$

where $c'$ is the true outcome for a particular patient. We propose inversely proportional normalised weights: $\sum_{c=1}^{C} w_c = 1$ and $w_c$ is inversely proportional to the class distribution, i.e., $w_c \propto \frac{N}{N_c}$ with $N$ being the number of patients and $N_c$ being the number of patients with outcome label $c$. Class weighting penalises misclassification more heavily on less sampled classes.

We also propose a novel distribution loss function, $L_{\text{dist}}(\boldsymbol{\pi})$. We define the average cluster probability of assignment as $\boldsymbol{\pi}_C := \frac{1}{N} \sum_n \boldsymbol{\pi}_n$. Then we introduce $L_{\text{dist}}(\boldsymbol{\pi}) = -H(\boldsymbol{\pi}_C)$, where $H$ denotes entropy. Note that $L_{\text{dist}}$ is minimised when $\pi_C$ is uniform, ensuring all clusters are 'explored' and have comparable number of samples. While clusters should not necessarily be assigned the same number of samples, this loss helps to overcome a concern of *cluster collapse*, where clusters do not separate and samples are assigned to a very small number of non-representative clusters. Finally, to separate cluster representation vectors, we define the cluster separation loss as $L_{\text{clus}}(\mathcal{C}) = -\frac{1}{K(K-1)} \sum_{i,j} \|\mathbf{c}_i - \mathbf{c}_j\|^2$

To optimise our model, iterative gradients are applied according to weighted combinations of the above loss functions with hyper-parameter weights $\alpha, \beta$:

1. Outcome Predictor is updated according to $L_{\text{pred}}$;
2. Encoder and Identifier are trained according to $L_{\text{pred}} + \alpha L_{\text{dist}}$;
3. Cluster representation vectors are updated with regards to $L_{\text{pred}} + \beta L_{\text{clus}}$.

## 3.6 Initialisation

Our proposed model also follows a set of initialisation pre-training procedures. Firstly, the Encoder and Outcome Predictor are pre-trained according to a classification task ($\tilde{\mathbf{y}} = P(E(\mathbf{x}))$), with corresponding loss $L_{\text{pred}}$. Latent state representations $E(\mathbf{x})$ are clustered through a K-means algorithm with $K$ clusters across the whole training set. Cluster representation vectors are initialised as given by the resulting cluster centroids, and finally the Cluster Identifier network is pre-trained to identify clusters as predicted by the K-Means algorithm with categorical cross-entropy loss. Implementation was completed in Python, with TensorFlow 2, scikit-learn and NumPy. All experiments were run with 1 Tesla v100 GPU, and 8 CPUs Intel(R) Xeon(R) Gold 6246 @ 3.30GHz.

## 4 Results

For Benchmark purposes, we considered TSKM as a classic clustering benchmark, and SOM-VAE and AC-TPC as state-of-the-art phenotypic clustering methods. AC-TPC considers temporal subsequences of a complete patient set of observations - for comparison purposes, we consider only the model output for the complete patient sequence. For simplicity, we present results with all input features considered.

All models were trained on the same training set (60% of the complete input data) and evaluated against the same test set (remaining 40%). For DL models, we further split the training set into a purely training and validation sets. All experiments with varying hyper-parameters were repeated 10 times with a fixed set of 10 distinct seeds, and results are reported according to average metric and standard deviation. A complete list of the hyper-parameters considered for each model is included in Table A7 in the Appendix. In bold, top-performing hyper-parameters are indicated. Optimal integer hyper-parameters ($K$, $l$) were selected according to an "Occam's Razor" approach - for each parameter, we assign it the highest value such that increasing this amount does not lead to a significant increase in performance according to mean AUROC and a Friedman's hypothesis test. Neural network size parameters were kept consistent across all DL models where applicable. All

other optimal hyper-parameters were selected according to highest AUROC performance conditional on the model predicting each class (e.g. not making predictions solely for no event or Death Events).

We evaluated clustering performance through standard clustering metrics, including Silhouette score (SIL, Rousseeuw (1987)), Davies-Bouldin Index (DBI, Davies (1979)), Variance Ratio Criterion (VRI, Caliński & Harabasz (1974)). Results for all clustering models are displayed in Table 3. In Table 4, we evaluated the (multi-class) prediction performance with regards to Area-under-the-Receiver-Operating-Curve (AUROC), unweighted mean F1-score, unweighted mean Recall, and Normalised Mutual Information (NMI). For purely unsupervised models (SOM-VAE and TSKM), an outcome predictive pipeline was constructed by assigning patient admissions to clusters, and consequently to the empirical outcome distribution in the corresponding cluster. The prediction task was also benchmarked against traditional classifiers for outcome prediction in EHR data, namely Support Vector Machines (SVM), XGBoost (XGB) and NEWS2 (i.e., the National Early Warning Score used in the UK hospitals). Furthermore, to evaluate other Neural Network models as benchmarks and also justify both proposed mechanisms, we furthermore considered two other benchmarks: a) ENC-PRED: a stacked LSTM Encoder, followed by a MLP network for outcome prediction, and b) ATTEP; a model equivalent to CAMELOT, except the original entropy loss is considered over the clustering dist loss. Supervised performance for all the above models is included in Table 4. As noted, convergence was particularly difficult for ATTEP due to cluster collapse.

| Metric | TSKM | SOM-VAE | AC-TPC | CAMELOT (proposed) |
|--------|------|---------|--------|---------------------|
| SIL | **0.35** (±**0.01**) | 0.25 (± 0.08) | 0.04 (± 0.01) | 0.11 (±0.04) |
| DBI | **1.19** (±**0.08**) | 1.89 (± 0.63) | 4.34 (± 0.80) | 3.12 (±0.53) |
| VRI | **554.6** (±**2.50**) | 12.8 (± 9.32) | 66.5 (± 18.7) | 216.7 (±6.2) |

Table 3: Clustering separability results by the different clustering methodologies given input data with all available features (static, vital-signs, serum and haematological variables). For each metric and model, the average score and standard deviation are returned. The best values for each metric are indicated in bold.

| Metric | AUROC | F1-score | Recall | NMI |
|--------|-------|----------|--------|-----|
| SVM | 0.50 (± 0.02) | 0.23 (± 0.00) | 0.25 (± 0.00) | 0.01 (± 0.02) |
| XGB | 0.65 (± 0.01) | 0.23 (± 0.00) | 0.22 (± 0.00) | 0.03 (± 0.04) |
| NEWS2 | 0.61 | 0.29 | 0.34 | 0.01 |
| TSKM | 0.55 (± 0.01) | 0.24 (± 0.03) | 0.26 (± 0.02) | 0.01 (± 0.03) |
| SOM-VAE | 0.61 (± 0.09) | 0.27 (± 0.05) | 0.27 (± 0.03) | 0.05 (± 0.03) |
| AC-TPC | 0.68 (± 0.01) | **0.38** (±**0.01**) | 0.36 (± 0.01) | 0.17 (± 0.02) |
| ENC-PRED | 0.57 (± 0.02) | 0.25 (±0.02) | 0.26 (± 0.02) | 0.06 (± 0.03) |
| ATTEP | 0.67 (± 0.02) | 0.36 (±0.02) | 0.36 (± 0.02) | 0.16 (± 0.03) |
| CAMELOT (proposed) | **0.73** (±**0.02**) | 0.36 (±0.01) | **0.38** (±**0.02**) | **0.20** (±**0.03**) |

Table 4: Outcome prediction scores across all models, displayed with an average and standard deviation of a set of 10 seeds (except NEWS2, which is deterministic). The best values for each metric are indicated in bold. For clustering algorithms, cluster outcome distributions were taken to be the empirically observed distribution in each cluster.

On top of performance evaluation with regards to clustering separability and outcome prediction, we display a comparison between the learnt cluster phenotypes of the proposed model and that of phenotypic clustering benchmark AC-TPC. For each cluster, the corresponding outcome propensity $P(\mathbf{c})$ is shown as a bar plot over the 4 possible outcomes with corresponding probability value. We also display cluster outcome propensity plots for both TSKM and SOM-VAE in the Appendix (Figures A.8 and A.9). We note that the cluster outcome propensity distributions learnt by CAMELOT also align with the empirical outcome relevance in the learnt clusters (Table A.12). Lastly, we also display feature-time cluster relevance attention maps in Figure 6. For each cluster, a patient was randomly selected from the set of patients in the corresponding cluster, and a corresponding feature-time attention matrix and visualised as a heatmap.

## 5 DISCUSSION

Our proposed model shows an improvement in clustering performance ( see Table 3) when compared to the current phenotypic clustering benchmark (AC-TPC), and outperforms SOM-VAE according

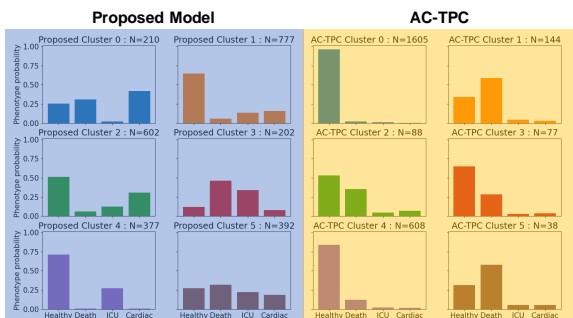

Figure 5: Comparison of bar plots of cluster outcome propensity distributions for the proposed model and benchmark AC-TPC. On the left (blue), distributions are displayed for each cluster (out of a total of 6), and each phenotype corresponds to the probability of an outcome. Similar results are shown on the right (yellow) for AC-TPC. The title of each sub-plot indicates the cluster considered, as well as the number of patients assigned to a given cluster.

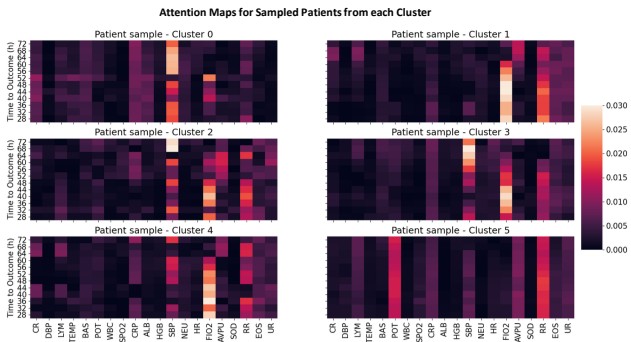

Figure 6: Feature-Time Cluster Relevance Map. Each Heatmap represents a feature-time relevance matrix for a random patient assigned to a given cluster. Vertical Axis indicates time to outcome, in hours, while horizontal axis indicates different input features.

to VRI. Although the cluster separability metrics are superior in the case of TSKM, this is expected given a metric bias towards convex clusters, and the convexity of the K-Means based algorithm. In particular, DL clustering occurs in a latent space, which, unfortunately, is not easily comparable with an algorithm targeting the input space (such as K-Means). Furthermore, we argue clusters learnt by TSKM are less relevant that our model's as a) TSKM clusters are extremely hard to distinguish with regards to outcome propensity (as evidenced by very low performance on a prediction task (Table 4)), and b) TSKM clusters are less separable with regards to trajectory evolution, as there is less separation of mean HR trajectories, and less cluster separation when data is projected to a two-dimensional domain with t-stochastic neighbour embedding (tSNE) - both figures are in the Appendix, Figures A10 and A11.

With regards to predictive power, it can be seen in Table 4 that our model outperforms both standard classifiers (at least 8%) and benchmarking clustering methods (at least 5%) according to mean AUROC. A similar increase can be seen in other classification metrics, with the exception of F1-score, where model performance is slightly below to that of AC-TPC. Our model is able to more accurately determine patterns in the data than the previously proposed models, as EHR data is extremely complex and heterogeneous. It is particularly promising that the model obtains good predictive task results despite a clustering bottleneck (i.e. sample predicted outcomes are done through the assigned cluster, as opposed to tailored to the precise input data). While it is possible that other models could show better performance on the direct task of outcome prediction given EHR input, such models can potentially be associated with lack of robustness or input sensitivity difficulties. Furthermore, it

is likely they would struggle with identifying relevant trends and properties of clinical interest. As such, for new admissions, such models could provide a prediction for the overall outcome, but no robust understanding of how this outcome will occur, and how to prevent potential risks of deterioration, let alone the ability to pool data from other similar patients.

Figure 5 shows the advantage of two key aspects of our methodology. Our model identifies clear, separable cluster outcome distribution and provides a useful layer of interpretability to clinicians to understand a potential risk of deterioration. Our model also identifies a more diverse set of cluster outcomes than AC-TPC, which only picks up 3 different cluster outcome distributions, and doesn't identify the presence of the "ICU" and "Cardiac" classes. We also show other cluster-phenotype benchmark results in Appendix A.8 and A.9. With regards to clusters learnt by CAMELOT, clusters $0$ and $3$ are the clusters with most ill cohort - they are largely representative of death and cardiac events on the subsequent $24$ hours. On the other hand, while cluster $2, 5$ are healthier, with a smaller chance of adverse events. Clusters $1$ and $4$ are largely "healthy" cluster, with reduced risks of the most intense adverse events. We note cluster outcome propensity distributions learnt by AC-TPC are unable to provide this level of detailed information. Furthermore, the propensity distribution learnt by our model matches with the empirical number outcome events observed in each cluster (displayed in Appendix Table A12). Note, furthermore, that the model managed to successfully navigate a heavy class-imbalance setting. Representative clusters are able to capture different-sized sub-populations, yet still identify potential risks of deterioration.

On the other hand, learnt cluster attention maps introduced in Figure 6 introduce yet another layer of interpretability to our proposed clustering model. The personalised attention maps highlight the relevant feature-time pairs driving patient cluster assignment, and can be used to identify the most important clinical variables. For instance, analysing Figure 5 suggests clusters with highest propensity for either of Death or Cardiac Arrest events are Cluster 0, 3 (and 2 to a slightly smaller extent). This reflects in the resulting attention maps, where SBP and FIO2 are highlighted as key clinical variables for cluster assignment. This conclusion is further corroborated when considering descriptive statistics of CAMELOT clusters (Table A13), as SBP and FIO2 are some of the few variables with some significant separation across clusters, and when considering trajectory evolutions (Figure A14). Lastly, note that feature-time weights are also relevant if potential deterioration events did not take place - so that we are more confident on a patient's health status. As an example, attention maps for clusters 1 and 4 (reasonably healthy clusters) indicate $FIO_2$ as very relevant towards the latter stages of the admission - this is likely due to these patients not showing an increase in oxygen intake (as they did not need it). Thus, attention maps can be very versatile.

## 6  CONCLUSION AND FUTURE WORK

In this work, we propose a novel deep learning model for the task of identification of phenotypically separable clusters applied to EHR data for. As part of our proposed model, we propose 2 distinct loss functions and introduce a novel feature-time attention layer to better represent patient data and to introduce a feature-time relevance map for each cluster. Our experiments show promising results with the addition of both methodological tools above, on both cluster separability and outcome prediction performance. The addition of the feature-time layer has the added benefit of introducing key interpretability tools for researchers to understand relevant regions for good patient physiology representation as well as an indication of what can lead to patient deterioration.

There are multiple interesting avenues of investigation building on this work. On the other hand, the current attention layer mechanisms could potentially be improved with the addition of temporal weight smoothness, or, alternatively, weight regularization to encourage exploration of the complete feature-time space. Alternatively, cluster selection through a neural network mechanism introduces high capacity at the potential cost of robustness and cluster collapse. Potentially, more traditional methods incorporated into a similarly complex pipeline can achieve better performing through a clearer identification of cluster regions in latent space. Furthermore, methodological improvements will also benefit from a more extensive testing across other diverse datasets and other potential areas of application.

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

# A APPENDIX

## A.1 DATA

A description of the complete pipeline of data re-processing, following the protocol defined in Pimentel et al. (2019), is shown in Figure A.1.

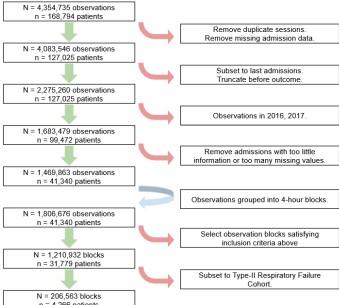

A.1: HAVEN processing

A total of 26 input features were considered. Firstly, 4-hourly vital-sign sets which included 8 features: Heart Rate (HR), Respiratory Rate (RR), Systolic Blood Pressure (SBP), Diastolic Blood Pressure (DBP), peripheral Oxygen Saturation ($SpO_2$), Temperature (TEMP), level of consciousness via the AVPU scale - Alert, Verbal, Pain, Unresponsive - and estimated Fraction of Inspired Oxygen ($FiO_2$, available when an oxygen mask is applied to the patient). Each set consisted of a timestamp and the vital-sign numerical values. Secondly, 4 demographic variables were selected (modelled as static variables): age, sex, and admission type (elective or surgical). Thirdly, we included 6 features resulting from biochemistry blood tests, denoted as 'Serum': Serum levels of urea, albumin, creatinine, sodium, potassium and C-reactive protein. Finally, 8 haematological blood test features were also included: white and haemaglobin cell counts, concentration of eosinophils, basophils, neutrophils, and lymphocytes, as well as eosinophil-to-basophil and neutrophil-to-lymphocyte ratios. These features were selected based on domain knowledge of features related to severity in the prognosis and outcome of T2RF inpatients.

Descriptive statistics for all input variables is described in Table A.2. Median and inter-quartile range (IQR) is displayed for continuous and categorical variables, while binary variables are shown according to number of counts in the dataset and corresponding cohort proportion. Statistics are displayed for the complete data ("All"), but also for each sub-cohort defined by the overall outcome. We can observe that these sub-cohorts are not clearly separable and are hard to identify solely from this information.

A summary of the patient cohort in relation to outcomes and target phenotypes can be seen in Table A.3. Challenges with regards to obtaining phenotypically separable clusters can similarly be observed - there is no clear significant difference between the target outcome sub-cohorts with regards to demographic input variables. With regards to outcome distribution, we also note the high degree of imbalance in the dataset - the large majority of the patients in our dataset suffered from no adverse events (over 86%), while only 48 had a Cardiac event, and 76 were re-directed to the ICU.

The lack of outcome sub-cohort separability can further observed in a temporal domain. Figures A.4, A.5, A.6 plot the mean trajectories of different temporal variables sets for each outcome sub-cohort, respectively, according to vital signs, haematological and serum features. Mean is calculated based on the time to outcome, and missing observations are disregarded and ignored.

| | Description | Units | Type | All | Cardiac | ICU | Death | Healthy |
|---|---|---|---|---|---|---|---|---|
| n | Patient Count | | Integer | 4266 | 48 | 76 | 441 | 3701 |
| N | Trajectory Count | | Integer | 110,916 | 1,248 | 1,976 | 11,466 | 96,226 |
| O | Observation Count | | Integer | 1,286,119 | 14,095 | 20,227 | 184,504 | 1,067,293 |
| **Vital signs** | | | | | | | | |
| HR | Heart-rate | beats/minute (bpm) | Integer | 82.50 (72 - 94) | 88 (77 - 100) | 88.50 (77 - 99) | 89 (77 - 100) | 81.67 (72 - 92) |
| RR | Respiratory-Rate | breaths/minute (Bpm) | Integer | 18 (16 - 19) | 18.29 (17.50 - 20) | 18 (16 - 19) | 19 (18 - 21) | 18 (16 - 18) |
| SBP | Systolic Blood Pressure | mmHg | Integer | 126 (112 - 141) | 122 (108 - 138) | 119 (104 - 137) | 123 (108 - 140) | 127 (113 - 142) |
| DBP | Diastolic Blood Pressure | mmHg | Continuous | 67 (60 - 76) | 65 (57 - 73) | 64 (57 - 72.33) | 66 (58 - 75) | 67 (60 - 76) |
| SPO2 | Estimated Oxygen Saturation | % | Continuous | 95 (94 - 97) | 95 (93 - 97) | 95 (94 - 97) | 94 (91 - 96) | 95 (94 - 97) |
| FIO2 | Fraction of Inspired Oxygen concentration | % | Continuous | 21 (21.00 - 28.67) | 21 (21 - 31) | 21 (21 - 41) | 28 (21 - 43) | 21 (21 - 21) |
| TEMP | Temperature | °C | Continuous | 36.40 (36.05 - 36.80) | 36.25 (36.00 - 36.60) | 36.40 (36.00 - 36.83) | 36.20 (36 - 36.65) | 36.40 (36.10 - 36.80) |
| AVPU | Alert, Verbal, Pain, Unresponsive Scale | | Categorical (1-4) | 1 (1 - 1) | 1 (1 - 1) | 1 (1 - 1) | 1 (1 - 1) | 1 (1 - 1) |
| **Static** | | | | | | | | |
| age | Patient age | year | Integer | 72 (62 - 81) | 76 (69 - 82) | 69 (61 - 80) | 81 (74 - 88) | 71 (61 - 74) |
| gender | Male patients | | | 2123 (49.77%) | 28 (58.33 %) | 38 (50.00%) | 247 (56.01 %) | 1810 (48.01 %) |
| Elective | Elective Admissions | | Binary | 1139 (26.70 %) | 2 (4.17%) | 8 (10.53 %) | 3 (0.68 %) | 1126 (30.42 %) |
| Surgical | Surgical admissions | | | 1130 (26.49 %) | 6 (12.50 %) | 22 (28.95%) | 48 (10.88 %) | 1054 (28.48 %) |
| **Serum** | | | | | | | | |
| HGB | Haemoglobin | g/L | Continuous | 11.20 (9.70 - 12.80) | 11.00 (9.35 - 12.40) | 10.35 (9.00 - 12.03) | 10.80 (9.30 - 12.60) | 11.30 (9.80 - 12.90) |
| WBC | White Blood Cell count (blood) | $\times 10^9$/L | | 10.03 (7.63 - 13.23) | 10.90 (8.41 - 15.36) | 10.41 (6.79 - 14.04) | 11.38 (8.34 - 15.82) | 9.80 (7.54 - 12.82) |
| EOS | EOSinophil count (blood) | $\times 10^9$/L | | 0.10 (0.02 - 0.22) | 0.06 (0.01 - 0.20) | 0.04 (0.00 - 0.15) | 0.03 (0.00 - 0.11) | 0.11 (0.03 - 0.24) |
| BAS | BASophil count (blood) | $\times 10^9$/L | Continuous | 0.04 (0.02 - 0.05) | 0.03 (0.02 - 0.06) | 0.03 (0.02 - 0.05) | 0.03 (0.02 - 0.05) | 0.04 (0.02 - 0.06) |
| EBR | Eosinophil-Basophil Ratio | | | 2.60 (0.75 - 5.50) | 1.50 (0.50 - 4.83) | 1.43 (0.17 - 4.50) | 1.00 (0.00 - 3.33) | 3.00 (1.00 - 5.86) |
| NEU | NEUtrophil count (blood) | $\times 10^9$/L | | 7.60 (5.34 - 10.70) | 8.88 (6.25 - 12.60) | 8.14 (4.73 - 11.25) | 9.31 (6.46 - 13.52) | 7.31 (5.19 - 10.19) |
| LYM | LYMphocyte count (blood) | $\times 10^9$/L | | 1.16 (0.77 - 1.68) | 1.00 (0.63 - 1.38) | 1.00 (0.55 - 1.65) | 0.84 (0.55 - 1.24) | 1.24 (0.85 - 1.75) |
| NLR | Neutrophil-Lymphocyte Ratio | | | 6.36 (3.76 - 11.67) | 9.40 (4.83 - 15.98) | 8.41 (4.33 - 15.55) | 10.84 (6.20 - 19.59) | 5.74 (3.51 - 10.11) |
| **Haematological** | | | | | | | | |
| ALB | ALBumin level (plasma) | g/L | | 26.00 (22.00 - 30.00) | 23.00 (20.00 - 28.00) | 23.00 (19.00 - 27.50) | 23.00 (19.00 - 28.00) | 27.00 (23.00 - 31.00) |
| CR | Creatinine level (plasma) | umol/L | | 77.00 (58.00 - 109.00) | 107.00 (78.75 - 154.25) | 78.00 (52.00 - 125.75) | 93.00 (63.00 - 145.00) | 74.00 (58.00 - 102.00) |
| CRP | C-Reactive Protein level (plasma) | mg/L | Continuous | 63.43 (21.30 - 137.58) | 51.50 (25.80 - 112.85) | 113.20 (32.45 - 226.05) | 85.20 (36.20 - 156.35) | 58.00 (18.68 - 131.33) |
| POT | POTassium level (plasma) | mmol/L | | 4.00 (3.60 - 4.40) | 4.20 (3.80 - 4.80) | 4.10 (3.70 - 4.50) | 4.10 (3.70 - 4.80) | 4.00 (3.60 - 4.30) |
| SOD | SODium level (plasma) | mmol/L | | 137.00 (134.00 - 140.00) | 136.00 (132.00 - 139.00) | 136.00 (133.00 - 139.00) | 138.00 (134.00 - 142.00) | 137.00 (133.00 - 140.00) |
| UR | UREa concentration levels | ml | | 6.40 (4.40 - 10.40) | 10.30 (6.45 - 18.15) | 7.05 (4.40 - 11.67) | 10.40 (6.40 - 17.20) | 5.90 (4.20 - 9.00) |

A.2: Descriptive statistics and information of all input data features. Variables are displayed with type, description, units and average statistics. We separate all features according to medical literature, including vital-sign, static, serum and haematological variables, and we also display statistics per outcome sub-cohort, defined as a cohort with those patients assigned to a given outcome.

| | No Event | Death | ICU | Cardiac |
|---|---|---|---|---|
| N | 3701 | 441 | 76 | 48 |
| Age (IQR) | 71 (61 - 80) | 81 (74 - 88) | 69 (61 - 74) | 76 (69 - 82) |
| Gender, M | 1810 (48.9%) | 247 (56.0%) | 38 (50.0%) | 28 (58.33%) |
| CCI (IQR) | 4 (3 - 13) | 14 (4 - 21) | 7 (4 - 17) | 15 (4 - 23) |
| Elective | 1126 (30.4%) | 3 (0.7%) | 8 (10.5%) | 2 (4.2%) |
| Surgical | 1054 (28.5%) | 48 (10.1%) | 22 (29.0%) | 6 (12.5%) |

A.3: Descriptive demographic variable information for each outcome sub-cohort.

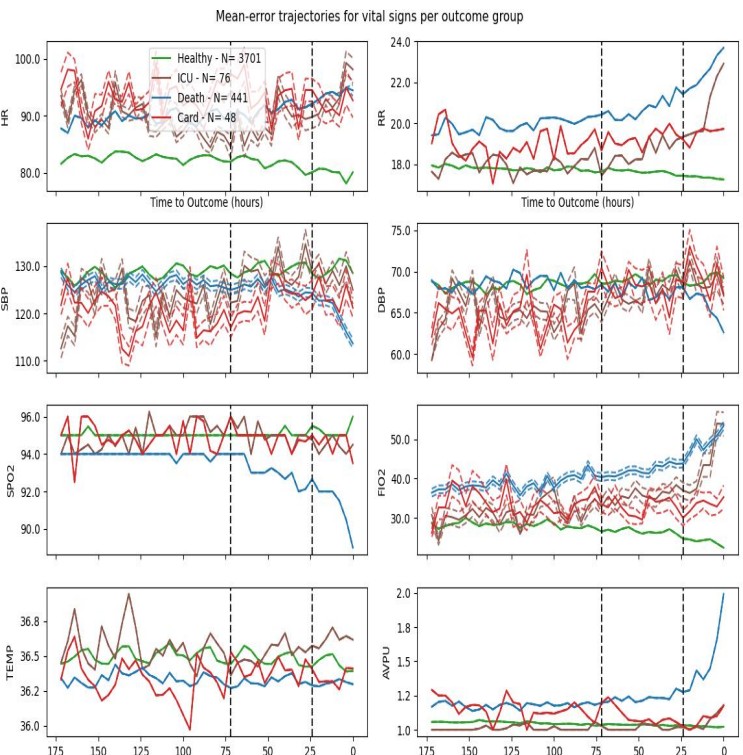

A.4: Plot of mean vital-sign trajectories (median with respect to $SpO_2$) in solid line as given by the 4 outcome groups: admissions with a) Cardiac-, b) Death- , or c) ICU-, and d) No-events. The respective standard errors are represented by the dashed lines. We visualised trajectories from up to 7 days prior to an outcome event or discharge - the black lines represent the time window (72 - 24 hours prior to an event or discharge) considered for input to all models.

## A.2  MODEL TRAINING

A list indicating the grid-search range of hyper-parameters considered in our experiments are indicated in Table A.7. For simplicity, we define $\mathbb{P} := \{0.001, 0.01, 0.1, 1, 10\}$, $\mathbb{L} := \{32, 64, 128, 256\}$ and $\mathbb{K} := \{3, ..., 20\}$. In bold, top-performing hyper-parameters according to target metrics defined in Section 4 are highlighted.

## A.3  RESULTS COMPARISON

In Figures A.8 and Figures A.9 we display cluster outcome propensity distributions for some of our experiments with benchmark clustering models SOM-VAE and TSKM, respectively. Both models do not naturally associate clusters with a distribution - we estimate the cluster outcome as the *empirical* outcome distribution for the patient cohort assigned to the corresponding cluster.

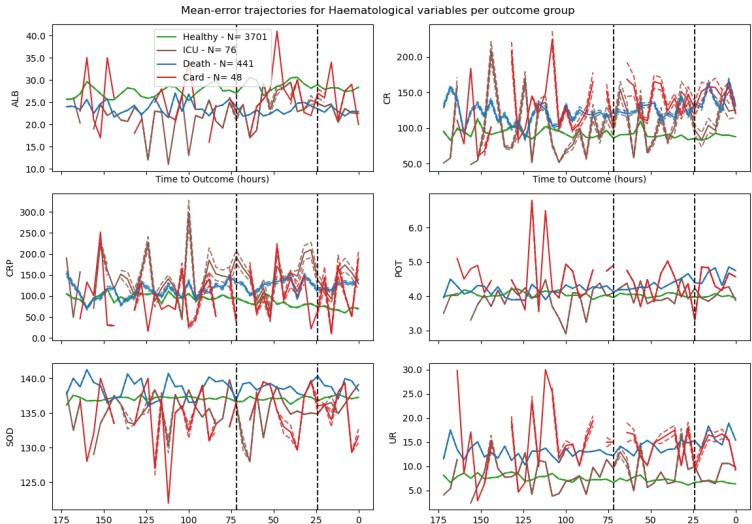

A.5: Plot of mean haematological trajectories in solid line as given by the 4 outcome groups: admissions with a) Cardiac-, b) Death-, or c) ICU-, and d) No-events. The respective standard errors are represented by the dashed lines. We visualised trajectories from up to 7 days prior to an outcome event or discharge - the black lines represent the time window (72 - 24 hours prior to an event or discharge) considered for input to all models.

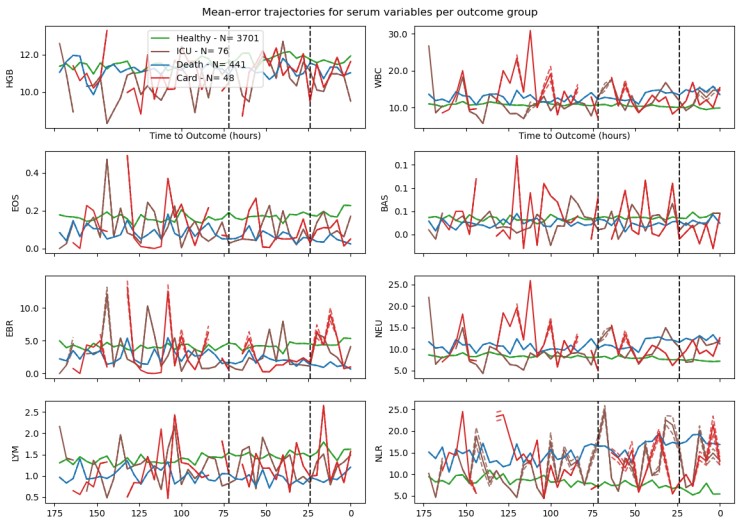

A.6: Plot of mean serum trajectories in solid line as given by the 4 outcome groups: admissions with a) Cardiac-, b) Death-, or c) ICU-, and d) No-events. The respective standard errors are represented by the dashed lines. We visualised trajectories from up to 7 days prior to an outcome event or discharge - the black lines represent the time window (72 - 24 hours prior to an event or discharge) considered for input to all models.

We note that clusters learnt by both clustering benchmark models have identical outcomes, which provides no useful clinical interpretability to the cluster-defined populations, as well as likely not assisting models to learn relevant cluster representations.

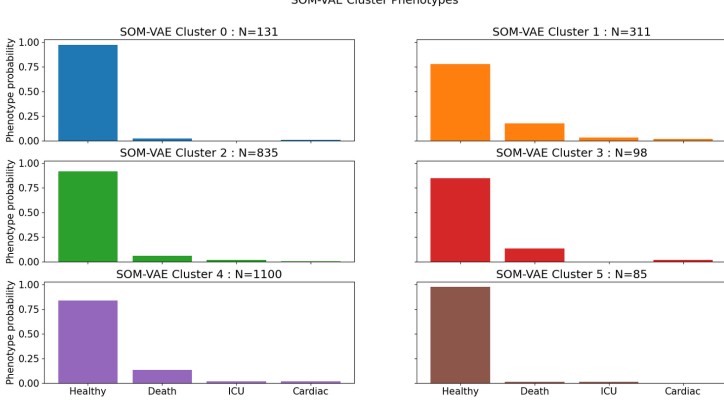

A.8: Bar plots of learnt cluster phenotypes for SOM-VAE with optimal hyper-parameters. Each plot represents a cluster - its phenotype is the corresponding empirical outcome distribution in its cluster-assigned patient cohort.

We go further in comparing clusters learnt by TSKM and by our proposed model. We argue clusters learnt by CAMELOT are much more relevant towards our goal. We show this through two distinct plots. Firstly, in Figure A10, we display a scatter plot of patients in each cluster (CAMELOT on the right and TSKM clusters on the left) after projection to two dimensions. Projection was completed through a principal component analysis reduction to 50 dimensions, followed by t-stochastic neighbour embedding dimensionality projection to two.

Furthermore, we also demonstrated that TSKM does not learn as separable cluster trajectory evolution profiles as CAMELOT. This is shown in Figure A11, where Heart-Rate mean trajectories for each cluster (i.e., average HR observations aligned to the same time until end of observations for patients in the clusters) are displayed. It is clear that CAMELOT cluster trajectories are easier to separate.

A complete description of the number of patient admissions with a given outcome per learnt cluster in our proposed model can be seen in Table A.12.

| Parameter | TSKM | SOM-VAE | AC-TPC | CAMELOT | SVM | XGB |
|---|---|---|---|---|---|---|
| seeds | $\{1001, 1012, 1134, 2475, 6138, 7415, 1663, 7205, 9253, 1782\}$ | | | | | |
| $\alpha$ | - | $\mathbb{P}(\mathbf{0.1})$ | $\mathbb{P}(\mathbf{0.01})$ | $\mathbb{P}(\mathbf{0.01})$ | - | - |
| $\beta$ | - | $\mathbb{P}(\mathbf{0.1})$ | $\mathbb{P}(\mathbf{0.01})$ | $\mathbb{P}(\mathbf{0.001})$ | - | - |
| $\gamma$ | - | - | - | - | - | $\{0.1, \mathbf{0.2}, 0.6\}$ |
| latent dim | - | $\mathbb{L}(\mathbf{64})$ | $\mathbb{L}(\mathbf{128})$ | $\mathbb{L}(\mathbf{128})$ | - | - |
| SOM dim | - | $\mathbb{L}^2(\mathbf{4}, \mathbf{4})$ | - | - | - | - |
| K | $\mathbb{K}(\mathbf{7})$ | - | $\mathbb{K}(\mathbf{6})$ | $\mathbb{K}(\mathbf{6})$ | - | - |
| kernel | $\{\mathbf{'DTW'}, \text{'euclidean'}\}$ | - | - | - | $\{\mathbf{'polynomial'}, \text{'rbf'}\}$ | - |
| C | - | - | - | - | $\mathbb{P}(\mathbf{10})$ | - |
| n-estimators | - | - | - | - | - | $\{100, \mathbf{200}, 300\}$ |
| depth | - | - | - | - | - | $\{1, 3, \mathbf{5}, 10\}$ |
| min-child-weight | - | - | - | - | - | $\{1, 2, 3, \mathbf{5}\}$ |

A.7: Parameter range used for Grid-search hyper-parameter optimisation. For each model, the list of parameter values tested is indicated. In **bold**, the optimum set of hyper-parameters is indicated for each model.

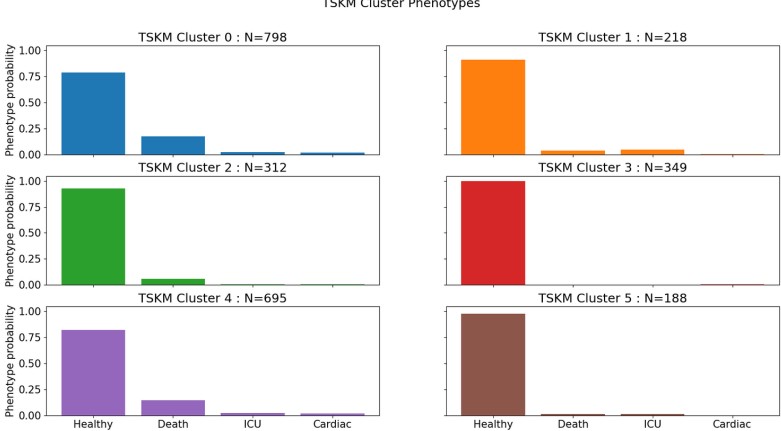

A.9: Bar plots of learnt cluster phenotypes for TSKM with $K = 6$. Each plot represents a cluster - its phenotype is the corresponding empirical outcome distribution in its cluster-assigned patient cohort.

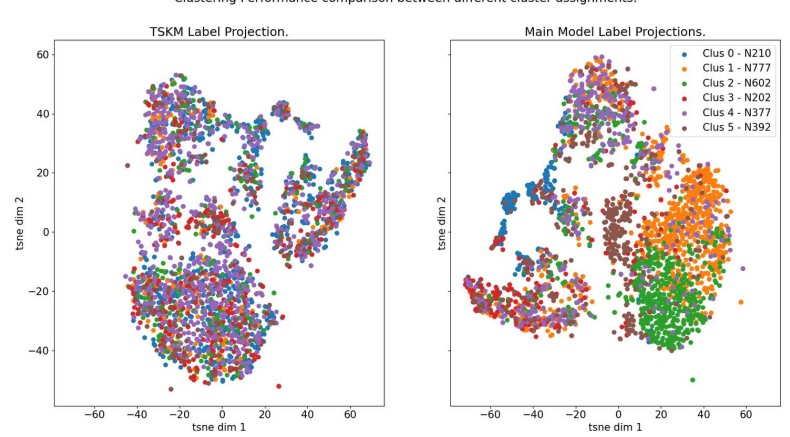

A.10: Scatter plot of cluster patient data after projection to 2 dimensions.

| Outcome | Healthy | Death | ICU | Cardiac |
|---------|---------|-------|-----|---------|
| Cluster 0 | 149 | 44 | 10 | 7 |
| Cluster 1 | 739 | 28 | 5 | 5 |
| Cluster 2 | 579 | 15 | 6 | 2 |
| Cluster 3 | 93 | 92 | 12 | 5 |
| Cluster 4 | 373 | 2 | 2 | 0 |
| Cluster 5 | 288 | 84 | 10 | 10 |

A.12: Table with empirical number of outcome admissions observed for each cluster learnt by the proposed model.

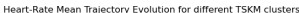

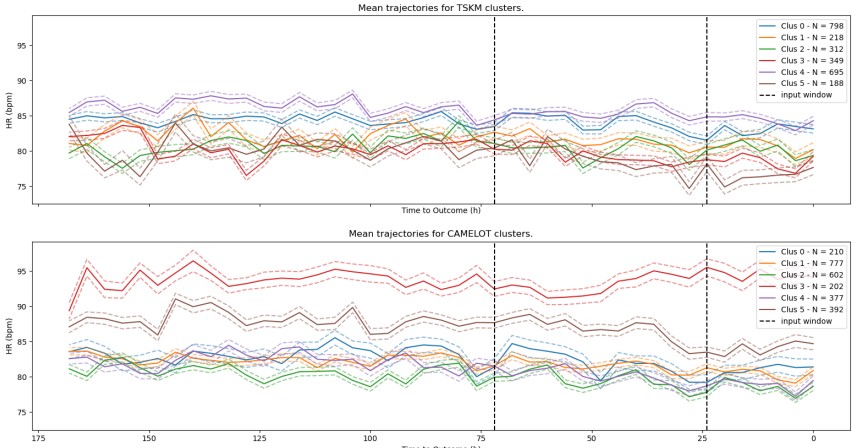

A.11: Plot of mean Heart-Rate (HR) trajectory in solid line as given by the TSKM learnt clusters (top) and CAMELOT (bottom). The respective standard errors are represented by the dashed lines. We visualised trajectories from up to 7 days prior to an outcome event or discharge - the black lines represent the time window (72 - 24 hours prior to an event or discharge) considered for input to all models.

We also computed summary statistics for the learnt CAMELOT clusters. For each of the resulting clusters, median, and quartile values were computed and plotted, except on the case of binary variables, where only the number of positive occurrences (and the corresponding proportion in the cluster) are shown.

Lastly, we plot the mean cluster trajectory evolution for SBP and FIO2 to present supportive evidence for the personalised attention maps in Figure 6. These two features were selected from attention map analysis.

| | Description | Units | Type | Clus 0 | Clus 1 | Clus 2 | Clus 3 | Clus 4 | Clus 5 |
|---|---|---|---|---|---|---|---|---|---|
| n | Patient Count | | Integer | 210 | 777 | 602 | 202 | 377 | 392 |
| **Vital signs** | | | | | | | | | |
| HR | Heart-rate | beats/minute (bpm) | | 82.0 (71.0 - 94.0) | 81.0 (72.0 - 91.0) | 80.0 (70.4 - 90.0) | 93.0 (83.0 - 103.0) | 82.0 (72.0 - 92.0) | 85.5 (74.0 - 97.0) |
| RR | Respiratory-Rate | breaths/minute (Bpm) | | 18.0 (16.5 - 19.0) | 18.0 (16.0 - 18.3) | 17.0 (16.0 - 18.0) | 19.3 (18.0 - 24.0) | 17.0 (16.0 - 18.0) | 18.0 (17.0 - 20.0) |
| SBP | Systolic Blood Pressure | mmHg | | 123.0 (107.9 - 142.0) | 136.0 (123.0 - 149.0) | 117.0 (106.3 - 130.0) | 130.0 (115.0 - 143.0) | 127.0 (114.0 - 141.0) | 123.0 (110.0 - 138.0) |
| DBP | Diastolic Blood Pressure | mmHg | Continuous | 65.0 (57.0 - 74.0) | 70.0 (63.0 - 78.7) | 65.0 (58.0 - 72.0) | 68.0 (60.0 - 78.0) | 67.0 (60.0 - 75.0) | 67.0 (59.0 - 75.0) |
| SPO2 | Estimated Oxygen Saturation | % | | 95.5 (94.0 - 97.0) | 95.0 (94.0 - 97.0) | 95.0 (94.0 - 97.0) | 94.0 (91.0 - 96.0) | 96.0 (94.0 - 97.0) | 94.0 (91.0 - 95.5) |
| FIO2 | Fraction of Inspired Oxygen concentration | % | | 21.0 (21.0 - 21.0) | 21.0 (21.0 - 21.0) | 21.0 (21.0 - 21.0) | 29.0 (21.0 - 54.0) | 21.0 (21.0 - 21.0) | 35.0 (21.0 - 49.0) |
| TEMP | Temperature | °C | | 36.3 (36.0 - 36.7) | 36.4 (36.1 - 36.8) | 36.3 (36.0 - 36.7) | 36.4 (36.1 - 36.9) | 36.5 (36.2 - 36.9) | 36.3 (36.0 - 36.7) |
| AVPU | Alert, Verbal, Pain, Unresponsive Scale | | Categorical (1-4) | 1.0 (1.0 - 1.0) | 1.0 (1.0 - 1.0) | 1.0 (1.0 - 1.0) | 1.0 (1.0 - 1.0) | 1.0 (1.0 - 1.0) | 1.0 (1.0 - 1.0) |
| **Static** | | | | | | | | | |
| age | Patient age | year | Integer | 78.0 (69.2 - 84.0) | 73.0 (63.0 - 81.0) | 71.0 (58.0 - 80.0) | 76.5 (69.0 - 85.0) | 70.0 (58.0 - 78.0) | 72.5 (64.0 - 81.0) |
| gender | Male patients | | | 145.0 (69.05 %) | 356.0 (45.82 %) | 301.0 (50.00 %) | 114.0 (56.44 %) | 196.0 (51.99 %) | 186.0 (47.45 %) |
| Elective | Elective Admissions | | Binary | 30.0 (14.29 %) | 210.0 (27.03 %) | 178.0 (29.57 %) | 8.0 (3.96 %) | 124.0 (32.89 %) | 110.0 (28.06 %) |
| Surgical | Surgical admissions | | | 40.0 (19.05 %) | 245.0 (31.53 %) | 201.0 (33.39 %) | 31.0 (15.35 %) | 99.0 (26.26 %) | 65.0 (16.58 %) |
| **Serum** | | | | | | | | | |
| HGB | Haemoglobin | g/L | | 9.9 (8.8 - 11.4) | 11.4 (10.0 - 13.0) | 11.4 (9.7 - 12.9) | 11.0 (9.5 - 12.7) | 11.1 (9.8 - 12.7) | 11.9 (10.4 - 13.2) |
| WBC | White Blood Cell count (blood) | x10^9/L | | 9.9 (7.4 - 13.5) | 10.0 (7.8 - 12.9) | 9.4 (7.1 - 12.4) | 12.5 (8.8 - 17.3) | 10.1 (7.8 - 12.9) | 10.5 (8.1 - 13.8) |
| EOS | EOSinophil count (blood) | x10^9/L | | 0.1 (0.0 - 0.2) | 0.1 (0.0 - 0.2) | 0.1 (0.0 - 0.2) | 0.0 (0.0 - 0.1) | 0.2 (0.1 - 0.4) | 0.1 (0.0 - 0.2) |
| BAS | BASophil count (blood) | x10^9/L | Continuous | 0.0 (0.0 - 0.1) | 0.0 (0.0 - 0.1) | 0.0 (0.0 - 0.1) | 0.0 (0.0 - 0.1) | 0.1 (0.0 - 0.1) | 0.0 (0.0 - 0.1) |
| EBR | Eosinophil-Basophil Ratio | | | 2.8 (0.8 - 5.8) | 2.3 (0.8 - 4.6) | 2.5 (0.9 - 5.4) | 0.8 (0.0 - 3.5) | 4.5 (2.3 - 7.6) | 2.0 (0.5 - 4.5) |
| NEU | NEUtrophil count (blood) | x10^9/L | | 7.6 (5.5 - 10.8) | 7.6 (5.4 - 10.5) | 6.9 (4.8 - 9.7) | 10.4 (6.6 - 14.6) | 7.2 (5.1 - 9.9) | 8.2 (6.0 - 11.5) |
| LYM | LYMphocyte count (blood) | x10^9/L | | 0.9 (0.6 - 1.4) | 1.2 (0.8 - 1.6) | 1.3 (0.9 - 1.8) | 0.9 (0.5 - 1.4) | 1.4 (1.0 - 2.0) | 1.0 (0.7 - 1.5) |
| NLR | Neutrophil-Lymphocyte Ratio | | | 8.7 (4.7 - 16.3) | 6.4 (3.8 - 10.6) | 5.3 (3.2 - 9.1) | 12.9 (6.4 - 22.3) | 4.8 (3.1 - 8.1) | 7.7 (4.8 - 13.4) |
| **Haematological** | | | | | | | | | |
| ALB | ALBumin level (plasma) | g/L | | 23.0 (18.0 - 28.0) | 27.0 (23.0 - 32.0) | 26.0 (22.0 - 30.0) | 24.0 (19.0 - 28.0) | 27.0 (23.8 - 31.0) | 26.0 (21.0 - 30.0) |
| CR | Creatinine level (plasma) | umol/L | | 210.0 (138.0 - 289.0) | 71.0 (58.0 - 91.0) | 72.0 (57.0 - 92.0) | 72.0 (48.0 - 105.0) | 71.0 (56.0 - 87.0) | 75.0 (56.0 - 107.0) |
| CRP | C-Reactive Protein level (plasma) | mg/L | Continuous | 71.8 (31.0 - 149.0) | 56.0 (17.1 - 136.4) | 47.0 (12.8 - 104.8) | 126.7 (57.8 - 213.7) | 63.5 (20.9 - 137.4) | 62.5 (25.6 - 138.1) |
| POT | POTassium level (plasma) | mmol/L | | 4.3 (3.9 - 5.0) | 4.0 (3.6 - 4.4) | 4.0 (3.6 - 4.3) | 3.9 (3.5 - 4.3) | 4.0 (3.6 - 4.3) | 4.1 (3.7 - 4.6) |
| SOD | SODium level (plasma) | mmol/L | | 136.0 (133.0 - 140.0) | 137.0 (134.0 - 140.0) | 137.0 (134.0 - 140.0) | 138.0 (135.0 - 142.0) | 138.0 (135.0 - 139.0) | 138.0 (134.0 - 140.0) |
| UR | URea concentration levels | mL | | 16.1 (10.7 - 23.6) | 5.8 (4.2 - 8.3) | 5.9 (4.2 - 8.7) | 7.3 (4.5 - 11.3) | 5.1 (3.7 - 6.8) | 6.9 (4.8 - 11.1) |

A.13: Descriptive statistics and information of all input data features. Variables are displayed with type, description, units and average statistics. We separate all features according to medical literature, including vital-sign, static, serum and haematological variables. Statistics are shown for each cohort as learnt by our model.

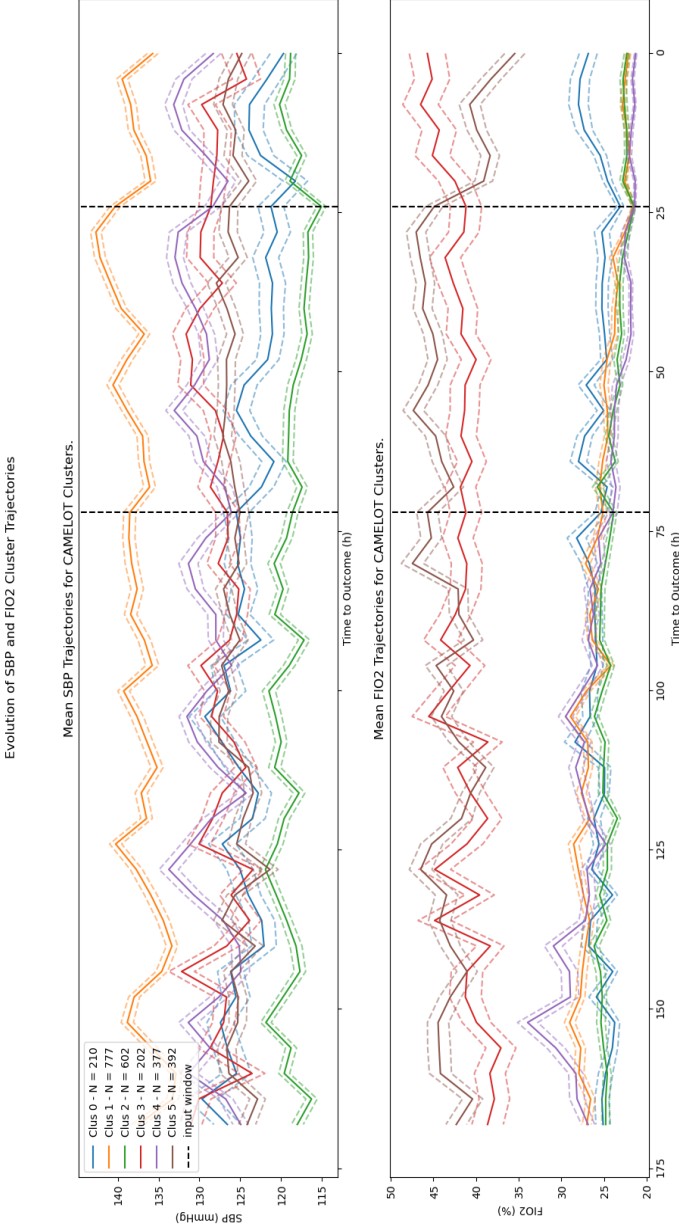

A.14: Plot of mean Systolic Blood Pressure (SBP) trajectories in solid line as given by the clusters learnt by our model (top). In the bottom, mean FIO2 trajectories are displayed. The respective standard errors are represented by the dashed lines. We visualised trajectories from up to 7 days prior to an outcome event or discharge - the black lines represent the time window (72 - 24 hours prior to an event or discharge) considered for input to all models.

