# OpenReview forum: "Cluster-based Feature Importance Learning for Electronic Health Record Time-series"
_ICLR.cc/2022/Conference — ICLR 2022 Submitted_

### Official Review · Reviewer_p3Vb · 2021-11-01

**Correctness:** 3
**Technical Novelty And Significance:** 2
**Empirical Novelty And Significance:** 2
**Recommendation:** 5
**Confidence:** 3

**Main Review:**

This paper tackles the important problem of phenotyping with an attention-based recurrent model. The neural network predicts an assignment for population-level latent centroids.

While the method is novel, it would be valuable to justify further the use of an LSTM, as attention-based mechanisms leverage long time dependencies while accelerating the training time of the model. The use of an LSTM might jeopardize this gain. A comparison with an MLP feature extractor would be valuable.

Further clarifications are necessary to describe the attention mechanisms (how is D_f chosen and what does it represent ?) and the model training (Why is the model trained iteratively ? It seems it could be done end to end).
In the experiment section, it is not clear why one should average performances over multiple seeds as it quantifies the modelling sensitivity to hyper parameter tuning and training, but not the model generalization to external datasets. Bootstrapped or cross-validation performances should be preferred. Additionally, the use of the 'Occam razor' selection of hyperparameters needs to be justified, as it is not clear to me how it is optimized in a large hyperparameter space.

In the results and discussion, metrics should be further described to highlight what they capture and what reflects a clinically relevant clustering. Figure 6 does not allow a better understanding of the clusters. It would be more interesting to show average attention for patients assigned to the same cluster.

The authors should also consider similar approaches proposed in the literature.
- Mechanisms that leverage both temporal and EHR data have been recently presented [1].
- Models which both cluster and model an outcome are also used in the statistical literature as profile regression models [2].
- Balanced loss is also widely used in the machine learning literature, the authors should not claim novelty on this part but cite papers such as [3]

Finally, a few minor revisions:
- "Traditional clustering models such as KMeans [...] have been shown to fail to capture the existing time dependent deature relationships" would need citations
- The authors describe a 4 hour moving window. Do they use separate window or overlapping ones, ie what is the frequency of the final data ?
- Median imputation is using validation and test data, shouldn't be training data ?



[1] Rocheteau, E., Liò, P. and Hyland, S., 2021, April. Temporal pointwise convolutional networks for length of stay prediction in the intensive care unit. In Proceedings of the Conference on Health, Inference, and Learning (pp. 58-68).

[2] Molitor, J., Papathomas, M., Jerrett, M. and Richardson, S., 2010. Bayesian profile regression with an application to the National Survey of Children's Health. Biostatistics, 11(3), pp.484-498.

[3] Cui, Y., Jia, M., Lin, T.Y., Song, Y. and Belongie, S., 2019. Class-balanced loss based on effective number of samples. In Proceedings of the IEEE/CVF conference on computer vision and pattern recognition (pp. 9268-9277).

**Summary Of The Paper:**

This work proposes a supervised approach to phenotype patients given their EHR trajectory and predicted outcome. The authors argue that the lack of interpretability of current deep learning approaches does not allow clinically relevant phenotypes and propose a feature-time attention mechanism to tackle this issue.


**Summary Of The Review:**

This work addresses an interesting and important problem in medical machine learning. However, it would benefit from additional methodological descriptions and results' interpretations.

---

> ### Author Response · Authors · 2021-11-23
> **Response to Reviewer p3Vb**
>
> We thank the reviewer for the time taken to review our manuscript and for the feedback provided. Given all the feedback indicated by all reviewers, we have made edits to the submission – all changes have been highlighted in yellow. Please also see above for a comment with a full description of the changes made.
>
>  **LSTM**: As discussed in the literature component of the work, there has been previous work completed through non-LSTM models to extract meaningful features from the input data (for instance, SOM-VAE, which leverages the power of CNNs). Given the resulting performance boost with ACTPC, it seems to be the case that leveraging long-term dependencies seems to improve input representation. Furthermore, LSTM networks naturally incorporate missingness through the inclusion of a mask, which allows inputs to be modelled more accurately, and with reduced imputation.
>
>  **Meaning of D_f**: We have added clarification into this variable in the manuscript– we use it to refer to the number of input features.
>
> **Iterative Model Training**: We train the cluster representations, Encoder-Identifier and Predictor iteratively, as they are trained with losses with different objectives, and each component is dependent on other network components. For instance, the Predictor takes cluster assignment (obtained by the Encoder-Identifier and the corresponding cluster representations) as an input and predicts the original patient outcome. On the other hand, cluster representations update optimise separation – which, for instance, makes the first task harder since cluster representations are now farther away.
>
>  **Parameter Selection**: We considered only two parameters to train following the Occam’s Razor principle – the dimensionality of the latent space (l) and number of clusters (K). We compute the average results over all pair combinations of (l, K). Consequently, we selected as “optimal” the minimum values of (l,K) such that increasing either does not result in a significant increase in AUROC performance gain. Our motivation behind this approach is that, on a theoretical level, it is possible to mimic results with a given value of (l, K) with any value (l’, K’) where l’ > l and K’>K.
>
>    *Attention Maps**: We sampled patient-specific attention to highlight that our model can provide interpretation in a personalized level. There is natural noise on the resulting attention maps, so that averaging results over clusters would lead to an approximation of the resulting cluster mean attention maps.
>
>     We thank the reviewer for making suggestions on other approaches. However, Ref [1] is mainly designed for a regression task in predicting hospital length of stay (albeit they consider addition of a binary mortality outcome) and benchmarks for our task include time-series clustering algorithms and multi-class prediction models. Ref [2] is a fully-Bayesian approach with MCMC, while our model proposes a different approach using deep learning without explicit model assumptions. Ref [3] is an interesting approach to deal with imbalanced classes for images. However, implementation of such an approach is not obvious for time-series healthcare data and it is beyond the scope of our work.
>
> We thank the reviewer for the remaining suggestions, as well. These have been added to the main text, when appropriate.
>
> References:
>
> [1] Rocheteau, E., Liò, P. and Hyland, S., 2021, April. Temporal pointwise convolutional networks for length of stay prediction in the intensive care unit. In Proceedings of the Conference on Health, Inference, and Learning (pp. 58-68).
>
> [2] Molitor, J., Papathomas, M., Jerrett, M. and Richardson, S., 2010. Bayesian profile regression with an application to the National Survey of Children's Health. Biostatistics, 11(3), pp.484-498.
>
> [3] Cui, Y., Jia, M., Lin, T.Y., Song, Y. and Belongie, S., 2019. Class-balanced loss based on effective number of samples. In Proceedings of the IEEE/CVF conference on computer vision and pattern recognition (pp. 9268-9277).

---

### Official Review · Reviewer_9DjJ · 2021-11-02

**Correctness:** 3
**Technical Novelty And Significance:** 1
**Empirical Novelty And Significance:** 2
**Recommendation:** 5
**Confidence:** 5

**Main Review:**

Strengths:
1) The authors did a good job in explaining and navigating the time-series complexity and challenges posed by EHR data particularly the limitations of previous work and the results section.
2) The paper provided comprehensive explanation of the proposed deep learning network especially the added model interpretability which plays a crucial role in integrating techniques/algorithms of artificial intelligence in healthcare.

Weakness:
1) My key concern about the paper is the lack of rigorous experimentation in designing the study cohort. Real-time diagnosis and prediction of clinical interventions is a major challenge especially using EHR data and for this study only the observations within 24 and 72 hours before the outcome were considered. The best practice while studying EHR-based predictions is to design an observation and prediction window, this practice helps understand how long the observations made by the model is valid and helps the clinicians to proactively assist their patients. The analysis presented in the manuscript lacks the experimentation with the sliding window and this further limits the scope for improvement and assessment of the models performance and determining the best possible subtype.

2) Similarly as noted above, the missingness treatment in the study also lacks experimentation. The foundation of the proposed deep learning framework is based on the quality of data and missing data is more prevalent in EHR but the authors failed to elaborate more on the missing data imputation section as in why authors chose to impute missing data with previous time block and median when there are other state-of-the art techniques/algorithms available to impute missing values especially when the quality improvement of the underlying data could increase cluster separability and model predictions. This could possibly help future readers of this paper to approach/reproduce/handle missing data problems in their respective healthcare settings.

3) The clustering separability of the proposed framework(CAMELOT) is significantly low than the benchmark TSKM and also the F-1 score of the outcome prediction scores is low, and AUROC is only marginally better than the other benchmark models. Unfortunately, the novelty in the method proposed by the authors does not reflect in the model outcomes.

Minor comments:
1) Authors should include what percentage of missingness was in the feature variables before imputation, perhaps in table A.2.
2) In section 3.6, I do not see citations/references for the libraries/software's used by the authors for this study. Please cite them appropriately.
3) I suggest authors choose a different color palette/schema to improve the interpretation of the feature-time cluster relevance map and make the heatmap(Figure 6) more visually appealing.
4) In Discussion section, 'learnt cluster attention maps introduced in Figure 6 introduce yet another layer' -> 'learnt

**Summary Of The Paper:**

The authors through this paper propose a deep learning framework to identify phenotypically separable clusters using EHR data and introduce a feature-time attention layer to better represent patient data including optimizing two loss functions to address class imbalance in the datasets. The study aims to make a more precision based clinical observation by leveraging feature dimensions and time-series nature of EHR data.

**Summary Of The Review:**

Although authors did a fairly good job in conceptualizing and developing the latter stages of their proposed deep learning framework, as I elaborated in my main review, the prior stages of the framework lack experimentation. Model-centric approach is important but data-centric approach is more crucial especially in healthcare where predictions play a vital role in assessing disease progression and therapeutic intervention.

---

> ### Author Response · Authors · 2021-11-23
> **Response to Reviewer 9DjJ**
>
> We thank the reviewer for the time taken to review our manuscript and for the feedback provided. Given all the feedback indicated by all reviewers, we have made edits to the submission – all changes have been highlighted in yellow. Please also see above for a comment with a full description of the changes made.  We address the particular concerns and questions raised above as follows:
>
> **Sliding Window**: The main goal of our work is to improve on existing methodologies on heterogeneous clinical trajectory data. While we agree time-window selection can be seen as another parameter for a modelling approach, we tried to leverage: a) Previous studies with the same clinical application, b) Clinical application of our methodologies and c) Data statistics on length of stay and average number of observations.
>
> Models such as NEWS2 are universally considered to make predictions with respect to 24 hour time window. As NEWS2 is an important clinical-driven benchmark, we similarly considered observations only up to 24 hours before any potential event. Similarly, research such as [1], [2] also consider an identical time window. This also matches clinical application - model output provides *new* information to clinicians, but also is wide enough for counter-intervention methods to take effect (e.g. medications).
>
> The upper bound of 72 hours was selected as it is close to the median value of length of stay for patient admissions. Our proposed model is not theoretically constrained by this upper bound as it is generic and can be used to identify clusters of arbitrary window lengths. While, performing a sliding window analysis is possible, it would likely lead to loss of generalisability due to overfitting and optimisation to this given dataset. Furthermore, selection of other thresholds would affect the input data in other ways, for instance, higher threshold would result in a smaller number of admissions (due to processing steps), while a lower threshold value would provide much less data for the model to train on.
>
>
>  **Missingness**: Similar to the point above, different approaches exist to work around missing values. As inputs were masked, however, we argue that difference in effects resulting from different imputation procedures are quite small. We do not wish to impute missingness with arbitrary algorithms as this is not used in a clinical setting. We mimic clinical practice by considering previously observed values when analysing the health status of a patient. We have kept the EHR data with minimum processing to ensure our model is reproducible in other hospitalised settings. Furthermore, in the clinical implementation aspect, we are demonstrating the generic contribution of our proposed model for identifying relevant clusters and phenotypes. Therefore, our model is versatile and can be used in any “processed” data and does not rely on the outputs of a specific data imputation algorithm.
>
> **Model Performance**: We argue that, despite inferior clustering performance than TSKM, our model learns more relevant cluster phenotypes. (Similar to answer to Reviewer nD3g, point 1).
>
> While the TSKM Baseline performs better when considering solely clustering metrics, we argue that the clusters learnt by TSKM are less relevant to our overall prediction task as well as performing worse at identifying separable cluster phenotypes.
>
> We clarify the definition of “phenotype” in Section 2 (highlighted in yellow) – a cluster phenotype is a combination of a) trajectory evolution profile, and b) characterisation with regards to an outcome of interest. In our setting, four patient outcomes were considered, based on admission events (or discharge). Outcomes are unknown for new admissions, yet they are ultimately what hospital clinicians are interested in identifying.
>
> TSKM Clusters are very hard to distinguish with regards to clinical outcome prediction. This can be seen in Table 4, where TSKM performs much worse than CAMELOT (0.55 AUROC for TSKM vs 0.73 AUROC). Furthermore, cluster trajectories learnt by TSKM are less separable with regards to trajectory evolution. We have added to the Appendix Figures A10 and A11 to illustrate this phenomenon, and to justify how pure standard clustering metrics, while useful, might fail to identify relevant feature trends in the input trajectories.
>
> We thank the reviewer for the suggestions, as well. These have been added to the updated draft for consideration. Thank you for the comments!
>
> REFERENCES:
>
> [1] Pimentel, Marco AF, et al. "Detecting deteriorating patients in hospital: development and validation of a novel scoring system." American Journal of Respiratory and Critical Care Medicine ja (2021).
>
> [2] F. E. Shamout, T. Zhu, P. Sharma, P. J. Watkinson and D. A. Clifton, "Deep Interpretable Early Warning System for the Detection of Clinical Deterioration," in IEEE Journal of Biomedical and Health Informatics, vol. 24, no. 2, pp. 437-446, Feb. 2020, doi: 10.1109/JBHI.2019.2937803.

---

> > ### Comment · Reviewer_9DjJ · 2021-11-29
> > **Response to authors**
> >
> > I want to thank the authors for clarifying my questions. I updated my score based on the discussion and revision.

---

### Official Review · Reviewer_MGut · 2021-11-05

**Correctness:** 3
**Technical Novelty And Significance:** 3
**Empirical Novelty And Significance:** 3
**Recommendation:** 6
**Confidence:** 3

**Main Review:**

Strengths: \
Writing is clear and well organized. \
Nice figures and appendix content. \
Good performance at prediction combined with cluster interpretability.


Cons: \
Only evaluated on private data. \
The clusters initialization. Is there a specific reason to use K-Means? Was it evaluated a different initialization method? \
The lack of ablation studies. The impact of removing or changing some of the framework's features, like the custom losses and Attention Block, should be evaluated to bring a stronger understanding of its impact on final performance. \
How does your Attention Encoder differ from RETAIN[1]? It looks very similar.


Minor comments:\
There is a missing blue dot and Figure 5



\
[1] Edward Choi, Mohammad Taha Bahadori, Joshua A Kulas, Andy Schuetz, Walter F Stewart, and Jimeng Sun. Retain: An interpretable predictive model for healthcare using reverse time attention mechanism. arXiv preprint arXiv:1608.05745, 2016.


**Summary Of The Paper:**

The paper describes a cluster-based learning method for EHR time series of in-hospital patients aimed to be interpretable and improve outcome prediction.\
It claims to contribute with an interpretable framework and its training process, with a weighted loss to deal with the class imbalance and a custom loss to avoid cluster collapse.

**Summary Of The Review:**

It is a good paper with relevant contributions.

I am not sure if the results are well evaluated so I expect to conclude at the discussion period.

---

> ### Author Response · Authors · 2021-11-23
> **Response to Reviewer MGut**
>
> We thank the reviewer for the time taken to review our manuscript and for the feedback provided. Given all the feedback indicated by all reviewers, we have made edits to the submission – all changes have been highlighted in yellow. Please also see above for a comment with a full description of the changes made.
>
> We address the concerns and questions raised above as follows:
>
> **Data Testing**: While we acknowledge the drawbacks resulting from testing on private data, it is, unfortunately, challenging to obtain access to healthcare datasets with temporal physiological information for secondary care data during ward stay. Most existing public datasets are based on ICU patients. These ICU patients are mostly in severe conditions under critical care, where they have a 1-to-1 nurse to patient ratio for monitoring deterioration.
>
> Here, we hope to help hospitalised patients staying in hospital wards where there is limited nursing/clinical staff and limited individualised attention to each patient. In these settings, phenotyping patient trajectories to predict their outcomes is more important.
>
> **Initialisation**: We initialised the cluster representations according to K-Means only. Other clustering initialisation methods can be applied, but we considered K-Means for two reasons:
>
>     a) Following available literature on time-series cluster learning ([3] and [4])
>
>     b) We are hopeful difference in cluster initialisation should not significantly impact learnt cluster phenotypes – representations are updated during training until convergence, and there is also a lack of consensus concerning an “optimal” clustering method for initialisation.
>
> **Ablation**: As mentioned in the answer to Reviewer nD3g, we have updated the Method section of the main paper to include a paragraph on ablation studies conducted. In effect, we have added Table W which shows the problem of cluster collapse on this imbalanced multi-class setting when we remove our proposed cluster loss. Similarly, we also evaluate our model if we substitute our Encoder by a regular LSTM Encoder network.
>
> **Comparison with Retain [1]**: While there are some similarities, our proposed Encoder differs from the architecture introduced in [1] in three key factors:
>
>      a) Firstly, mechanisms in the RETAIN architecture follow a combination of weighted-combinations and dense layers (correspondingly, with parameters to be learnt) and all parameters are backpropagated through the output prediction loss. In our case, parameters are learnt through approximation to relevant latent features.
>
>      b) Secondly, the RETAIN architecture applies a temporal-invariant input transformation to convert inputs into embeddings v. While this approach is possible, obtaining relevant time-specific feature importance information is therefore much more challenging, as the RNNs (non-linear maps) are acting on a transformed input. Our model follows a simpler approach: on a smaller parameter space, latent representations are directly “deconstructed” into information from each input feature. Furthermore, this is not time-invariant, which allows for temporal shift of feature relevance weights.
>
>      c) Thirdly, RETAIN does not consider a clustering setting. Our proposed method not only allows for obtaining feature-time relevance values, but also to compute this with regards to *each* cluster, therefore providing yet another interpretability to understand the existing clusters, and to map this relevance to raw input space (feature-time).
>
> We thank the reviewer for the suggestions, as well. These have been added to the updated draft for consideration. Thank you for the comments!
>
> REFERENCES:
>
> [1] Pimentel, Marco AF, et al. "Detecting deteriorating patients in hospital: development and validation of a novel scoring system." American Journal of Respiratory and Critical Care Medicine ja (2021).
>
> [2] F. E. Shamout, T. Zhu, P. Sharma, P. J. Watkinson and D. A. Clifton, "Deep Interpretable Early Warning System for the Detection of Clinical Deterioration," in IEEE Journal of Biomedical and Health Informatics, vol. 24, no. 2, pp. 437-446, Feb. 2020, doi: 10.1109/JBHI.2019.2937803.
>
> [3] Lee, Changhee, and Mihaela Van Der Schaar. "Temporal phenotyping using deep predictive clustering of disease progression." International Conference on Machine Learning. PMLR, 2020.
>
> [4] Baytas, Inci M., et al. "Patient subtyping via time-aware LSTM networks." Proceedings of the 23rd ACM SIGKDD international conference on knowledge discovery and data mining. 2017.

---

> > ### Comment · Reviewer_MGut · 2021-11-29
> > **Response to authors**
> >
> > Thank you authors for clarifying my questions and addressing my considerations.
> > I have updated my novelty score based on the discussion and the revised manuscript.

---

### Official Review · Reviewer_nD3g · 2021-11-08

**Correctness:** 3
**Technical Novelty And Significance:** 3
**Empirical Novelty And Significance:** 2
**Recommendation:** 8
**Confidence:** 4

**Main Review:**

Strengths:
1. The paper introduces time-feature relevance maps, which enable domain experts to look at features-time combinations which are in some important for their predictions.
2. The paper is written in a manner which makes experiments fairly easy to reproduce.
3. The proposed method is better than the National Early Warning Score used in the UK hospitals.

Weaknesses/Scope for improvement:
1. The TSKM baseline performs really well for the clustering problem. The authors claim that the superior results of TSKM are due to the metric bias towards convex clusters. However, given the capacity of deep neural networks and the ease of learning convex clusters, I am not convinced why the deep learning method would be at a disadvantage. The authors also claim that clustering on the input space and latent space cannot be compared. However, my understanding is that learning a smaller dimensional latent representation should instead help clustering, if the latent representations are meaningful.
2. I believe that in addition to the distribution of likely outcomes, common characteristics of patients in a cluster would be clinically relevant. For example, the could the authors come up with a way to use their proposed feature-time relevance maps to characterize each phenotype? I believe that would be very helpful.
3. There are no experiments which serve as evidence that the proposed loss functions are (1) needed, and (2) work. I can imagine that the authors can design simple experiments to illustrate the cluster collapse problem etc. which their loss functions are designed to handle. These experiments would both establish the need and serve as a "proof" that the loss functions are working as expected.
4. I am not sure if the baselines to predict outcomes directly from EHR data are strong enough. I suggest using stronger baselines to directly predict outcomes, for example 1-D ResNets have been found to be really performant time series classification models.

Questions:
1. Are the cluster representations $\mathcal{C}$ interpretable?
2. The authors are phenotyping patients based on unique distributions of outcomes. In this respect, how does one define "clearer" and "more separable" cluster phenotypes? More generally, how does one define a "good" phenotype? I feel the distributions in case of AC-TPC model are "clear" too.
3. I am not sure how does the proposed model handle different features having different sampling frequencies.

Suggestions:
1. The figures do not seem to have high quality. I recommend saving them as .svg or .pdf to ensure the same.
2. Typographical/grammatical errors:
     -- "which can has a well-known solution" => "which has a well-known solution"
     -- "with a smaller change of death" => "with a smaller chance of death"
3. Appendix A.10 has the true number of patients per outcome for each cluster. It would be interesting to see how close is the distribution of outcomes to the ground truth, based on a measure of distance between distributions.
4. Ensure that the colour maps in Fig.6 have the same range i.e. the same mapping between the colour and the relevance value.

**Summary Of The Paper:**

The paper presents a way to predict outcomes of patients using EHR records. The authors propose a phenotyping model which clusters similar patients based on their outcome distribution. They also introduce feature-time relevance map which helps explain for a particular patient the feature-time combinations which are important for their prediction.

**Summary Of The Review:**

I believe that the work is interesting, solves an important problem and does well. However, further experiments are needed, for instance, to check why the model is not performing as well in the clustering problem.

---

> ### Author Response · Authors · 2021-11-23
> **Response to Reviewer nD3g**
>
> We thank the reviewer for the time taken to review our manuscript and for the feedback provided. Given all the feedback indicated by all reviewers, we have made edits to the submission – all changes have been highlighted in yellow. Please also see above for a comment with a full description of the changes made to the overall manuscript. We address the concerns and questions raised above as follows:
>
> **TSKM Baseline comparison**: While the TSKM Baseline performs better when considering solely clustering metrics, we argue that the clusters learnt by TSKM are less relevant to our overall prediction task as well as performing worse at identifying separable cluster phenotypes.
>
> We clarify the definition of “phenotype” in Section 2 (highlighted in yellow) – a cluster phenotype is a combination of a) trajectory evolution profile, and b) characterisation with regards to an outcome of interest. In our setting, four patient outcomes were considered, based on admission events (or discharge). Outcomes are unknown for new admissions, yet they are ultimately what hospital clinicians are interested in identifying.
>
> TSKM Clusters are very hard to distinguish with regards to clinical outcome prediction. This can be seen in Table 4, where TSKM performs much worse than CAMELOT (0.55 AUROC for TSKM vs 0.73 AUROC). Furthermore, cluster trajectories learnt by TSKM are less separable with regards to trajectory evolution. We have added to the Appendix Figures A10 and A11 to illustrate this phenomenon, and to justify how pure standard clustering metrics, while useful, might fail to identify relevant feature trends in the input trajectories.
>
> **Cluster Descriptions**: We have updated our results and discussion section to include a more comprehensive description of each cluster. We have also added a table of summary statistics for clinical variables (e.g., demographics) over each of the learnt clusters - these further support/corroborate the description obtained from analysis of results obtained in Figure 5 and Figure 6. This new table is in the Appendix in Figure A13.
>
>  **Ablation**: We have added a paragraph to the Methods section indicating additional experiments conducted to verify the usefulness of our proposed modifications. In summary, we conducted similar experiments substituting our clustering loss with the original entropy loss of AC-TPC (ATTEP), and we also considered a standard LSTM Encoder followed by a Predictor block (LSTMEP). The former performed similar to AC-TPC (0.68 AUROC), but the latter performs drastically worse (0.57 AUROC).
>
>  **Benchmarks**: LSTMEP is equivalent to Encoder-Predictor initialisation (i.e. CAMELOT without clustering) and was consequently added to our benchmarks.  It can be seen that the addition of the clustering mechanism helps to improve outcome predictive performance, thus providing more support in terms of ablation. Similarly, the addition of the Attention Encoder allowed AUROC performance to increase from 0.68 to 0.73.
>
> **Concerning the questions raised**:
>
>     1. Are the cluster representations interpretable?
> Here, C is a vectorised form of the clustered latent representation z, therefore is not directly relevant for interpretability. This motivated the introduction of the cluster-based attention feature map to link to the raw input variables and to make it interpretable.
>
>     2. The authors are phenotyping patients based on unique distributions of outcomes. In this respect, how does one define "clearer" and "more separable" cluster phenotypes? More generally, how does one define a "good" phenotype? I feel the distributions in case of AC-TPC model are "clear" too.
>
> Further to the clarification of a definition for phenotype in the new draft, we claim the clusters learnt by AC-TPC (state of the art benchmark) are
>
> a) less separable/distinct due to the existence of multiple  pairs of clusters with a very similar outcome phenotype (for e.g., AC-TPC Clusters 0 and 4, 1 and 5, 2 and 3).
>
> b) less clear, as the outcome cluster phenotypes are only able to identify two outcomes out of four existing possible outcomes.
>
>      3. I am not sure how does the proposed model handle different features having different sampling frequencies.
>
>     For all time features, we aggregate feature observation values over one-hour blocks. All time features are consequently imputed following: a) value-carry forward, b) value-carry backward (useful for information missing at the beginning of the admission) and c) median value imputation.
>
> Frequencies with smaller sampling frequencies are less heterogeneous over an admission due to a higher amount of missingness and the above imputing procedure. The Encoder component of the network should be able to identify that such features vary less over time, and model it internally accordingly.
>
> We thank the reviewer for the suggestions, as well. These have been added to the updated draft for consideration. Thank you for the comments!

---

> > ### Comment · Reviewer_nD3g · 2021-11-29
> > **Response to author comments**
> >
> > I would like to thank the authors for conducting additional experiments and making changes to their manuscript. I have updated my score based on the revised manuscript.

---

### Author Response · Authors · 2021-11-23
**Manuscript Edits**

Thank you all the reviewers for the feedback provided and for the time taken to review our manuscript. We would like to bring to your attention changes to the manuscript that resulted from this feedback. All changes are highlighted in yellow, and, at a high level, they are as follows:

**Clarification on Phenotype Definition**: a cluster phenotype comprises of both trajectory evolution profile (how features vary over time) and an outcome propensity distribution. This has been added to Section 2), and some sentences have been re-phrased to clarify which component of the phenotype is being discussed at a given time.

**Clustering Performance**: We have added a paragraph to the discussion section, as well as added Figures A10 and A11 in the Appendix to argue that TSKM learnt clusters are not useful for our task. We have added Table A13 in the Appendix to provide some further clinical characterisation of learnt clusters by our model.

**Ablation**: We have added experiments to introduce further baselines to the supervised prediction task with two models considered:

a) An LSTM Encoder, followed by a feed forward neural network for multi-class prediction (LSTMEP).

b) Substituting our proposed cluster loss by an entropy loss, similar to [1] (ATTEP).

**Figure and Discussion Update**: We have updated the figures in the results section for clarity and for updating results obtained. We have also expanded the Discussion Section to clarify how the results obtained (phenotypes and attention maps) can be leveraged to inform clinicians, while we support these claims with new material in the Appendix.

We hope the responses above, and to each individual comment, together with the revised draft have addressed the concerns raised. Thank you once again!!

REFERENCES:

[1] Lee, Changhee, and Mihaela Van Der Schaar. "Temporal phenotyping using deep predictive clustering of disease progression." International Conference on Machine Learning. PMLR, 2020.

---

### Decision · Program_Chairs · 2022-01-20

**Decision:**

Reject

**Comment:**

This paper has been independently assessed by four expert reviewers. Two of them recommend acceptance (one straight, one marginal), and two rejection (both marginal). Among the main limitations of the presented work, found by the reviewers, was the limited reproducibility of the results due to the use of private data. The authors attempted to defend their experimental design choice to use only one private dataset by stating difficulty in obtaining public data that would be suitable for their method. I am personally in a strong disagreement with that statement, and with the sub-statement of the authors that some publicly available ICU data may not be suitable. That either signals limited practical utility of the presented approach to non-ICU settings only, or is simply incorrect. The presented approach nonetheless has an intriguing potential and I would be inclined to recommends its acceptance. Alas, I find the lack of reproducibility to be a significant drawback of the way this work is currently presented and this limitation could not be easily resolved. Therefore I am leaning towards recommending a rejection.